# Cannabidiol in Dentistry: A Scoping Review

**DOI:** 10.3390/dj10100193

**Published:** 2022-10-17

**Authors:** Carla David, Alejandro Elizalde-Hernández, Andressa S. Barboza, Gabriela C. Cardoso, Mateus B. F. Santos, Rafael R. Moraes

**Affiliations:** 1Biopathological Research Group, Faculty of Dentistry (GIBFO), University of the Andes Mérida, Mérida 5101, Venezuela; 2Graduate Program in Dentistry, Universidade Federal de Pelotas, Pelotas 96015-560, Brazil

**Keywords:** plants, cannabinoids, cannabis, dental medicine, patents

## Abstract

Cannabidiol (CBD) has been gaining increased attention in contemporary society but seems to have been little explored in dentistry. This scoping review mapped the scientific and technological scenarios related to the use of CBD in dentistry. Peer-reviewed publications were searched in five international databases, patents were searched in five technological platforms. In total, 11 articles and 13 patents involving CBD in dentistry-related applications were included. The countries contributing to most articles were Brazil (27.3%) and USA (18.2%). The studies involved experiments on animals (63.6%) and/or using bacteria or cells (36.4%), and no clinical study was found. Three different applications of CBD were observed: periodontal therapy (45.4%), aid for bone regeneration (27.3%), and general use in oral therapies (27.3%). Patent inventors were based in China (53.8%) or USA (46.2%). The patent claims were mainly compositions for oral care, tooth whitening, injury repair, antifungal, anti-inflammatory, and analgesic effects. A total of 76.9% of the patents were filed in association with a company. In general, research suggests that CBD has promising biological properties for applications in dentistry, whereas patents indicate that the current interest of industry relies on compositions for oral care. There appears to be extensive room available for research and technological applications of CBD in dentistry.

## 1. Introduction

Endocannabinoids are naturally occurring molecules in mammals produced in the endocannabinoid system [1,2], which interact with endogenous substances (e.g., anandamide and 2-arachidonoylglycerol), phytocannabinoids (e.g., cannabidiol, tetrahydrocannabinol), or synthetic cannabinoid analogs [3]. Cannabinoids can bind to specific CB1 and CB2 receptors found in plasma membranes, nerve fibers (CB1), and tissue cells (CB2) [1]. CB1 receptors in the brain are predominantly presynaptic and responsible for regulating memory, mood, sleep, appetite, and pain by releasing neurotransmitters. CB1 receptors are also present at lower concentrations in peripheral tissues, including cardiac, testicular, muscular, hepatic, pancreatic, and adipose tissues. CB2 receptors are probably responsible for the immunomodulatory and anti-inflammatory effects of cannabinoids and are expressed in the spleen and hematopoietic cells [2].

Cannabidiol (CBD) is a compound without psychoactive activity derived from Cannabis sativa plant [4]. CBD has been investigated more frequently and to a greater extent when compared with other cannabinoids in the contemporary literature [4], and has shown potential as a therapeutic agent in various physiopathological conditions or disease states [4]. Therapeutic uses of CBD have been linked to its anti-inflammatory [5], antioxidant [2], and analgesic [6] properties, including varied applications such as in bone tissue cell differentiation [3,7,8,9], neuroprotection, antiepileptic, anxiolytic, and anti-cancer agents [10]. CBD may also be used for health benefits that are not yet thoroughly studied, including stress relief, relaxation, and sleep improvement. The product can be used in edible, tincture, and vape modes.

Increasing research and development of CBD and its medical applications have been undertaken due to financial support by various private and governmental organizations in the past years. The global CBD market is expected to reach approximately USD 30 billion by 2025 [11]. CBD has a variety of potential uses in dentistry and oral medicine. Research targeting the therapeutic possibilities of CBD has been developed relative to its use in the oral cavity [12] and expression of endocannabinoid receptors in tooth, periodontal, and bone tissues [13,14], for instance. Therefore, it would appear to be promising to explore the possibilities for future research on the use of CBD, its compounds, or their synthetic analogs for dental applications. The aim of this scoping review was to map the contemporary scientific and technological scenarios related to the use and applications of CBD in dentistry at present.

## 2. Materials and Methods

This scoping review has been reported according to the Preferred Reporting Items for Systematic Reviews and Meta-Analysis Extension for Scoping Reviews guideline [15]. The study protocol was based on the framework proposed by Peters et al. (2015) [16], according to The Joana Briggs Institute [17], and is available at https://osf.io/yk65c/ (accessed on 10 October 2022). For the mapping, the following parameters were selected: (i) Population—articles and technological products; (ii) intervention—use of CBD or its synthetic analogs; (iii) comparison: other substances or treatments that may have been tested (if applicable); (iv) dental and oral medicine applications, (v): articles or patents. One general question guided the review: What scientific applications and technological products based on CBD or its synthetic analogs are being used in dentistry at present?

### 2.1. Eligibility Criteria

The inclusion criteria consisted of in vitro or in vivo studies and patents that have evaluated or reported the use of CBD for dental applications or subjects (e.g., periodontology, oral surgery). Studies that evaluated the use of CBD alone or associated with other substances (e.g., vitamins) or biomaterials in dentistry were included. In addition, studies that evaluated other synthetic analogs (HU-308, HUF-101, fluoro-cannabidiol, HU-320, CBD-DMH-7-oic acid, deoxy-CBD, 11-hydroxy-CBD or O-1602) were also included. The search was restricted to documents published in English without restriction on date. Case reports, case series, pilot studies, opinion articles, letters, and conference abstracts were excluded due to the high risk of bias associated to these articles. Reviews were excluded to concentrate the analysis to original articles. Studies on the use or smoke of cannabis plants and other cannabinoids also were excluded. Patents including CBD were excluded when its application or scope of patents was not related to dentistry or oral medicine.

### 2.2. Information Sources and Search

The searches in literature databases were performed by two independent reviewers (CD and AE-H) and patent tools (CD and AB), with no starting date and continuing through to December 2021. The research for articles was carried out in five international databases: PubMed/MEDLINE, Scopus, Web of Science, Embase, and Cochrane Library. The search strategy was based on the MeSH terms of PubMed and the specific terms of the other databases, using the search strategies presented in Table 1. The reviewers also hand-searched the reference lists of the included articles to identify additional manuscripts. In addition, a search and analysis of patent applications was conducted via the following databases—United States Patent and Trademark Office (USPTO), Google Patents, World Intellectual Property Organization (WIPO), Espacenet (European Patent Office, EPO), and Questel Orbit (Paris, France), which contains patent data on over 90 authorities. The search strategy is described in Table 1. Furthermore, a patent search was carried out using the following International Patent Classification codes: A61K-6/00 (preparations to dentistry), A61K-8/97 (derived to algae, fungi, lichens or plants; from derivatives thereof). The searches were conducted in December 2021.

### 2.3. Selection of Sources of Evidence

All records identified were imported into the EndNote program (EndNote X9; Thomson Reuters, New York, NY, USA). The independent researchers identified articles and patents by first analyzing titles and abstracts for relevance and eligibility criteria using the online system Rayyan QCRI (Hamad Bin Khalifa University, Doha, Qatar). Retrieved records were classified as included, excluded, or uncertain. Full-text versions of the included and uncertain records were selected for further eligibility screening. Discrepancies in a screening of titles/abstracts and full texts were resolved by discussion. In case of disagreement, the opinion of a third reviewer (RRM) was sought.

### 2.4. Data Charting, Data Items, and Analysis

We created two spreadsheets in the Microsoft Office Excel 2013 software (Microsoft Corporation, Redmond, Washington, DC, USA) for data extraction from articles and patents. The spreadsheets were pilot tested by three reviewers (CD, AB, RRM) to reach a consensus on which data to collect and how. Two independent reviewers (CD, AB) extracted the main relevant data, specifically focusing on the outcomes in dentistry, and influence of CBD on the dental applications. The following items of data were collected for articles: year, first author, journal, study design, country of the corresponding author, CBD dosage, route of administration, CBD presentation, manufacturers, applications (e.g., periodontal therapy, aid for bone regeneration, oral therapy), and sponsors of the studies. From patents, the following data were collected: patent numbers, year, title, inventors, country of inventors, main claims, and company. The mapping of research findings and patents was carried out considering the application of CBD in different dental subjects.

## 3. Results

A total of 2312 unique articles and 989 patents were identified (Figure 1) with 2294 articles and 974 patents excluded based on title and/or abstracts. From the 18 full-text articles and 15 patents assessed for eligibility, 11 studies and 13 patents were included in the qualitative synthesis, making up a total of 24 documents included in this review.

### 3.1. Characteristics of Studies

Characteristics of the 11 original studies included are presented in Table 2. The studies were published between 2009 and 2020 in various journals, the majority of which could not be categorized as being dedicated to publishing dental science only. The countries that contributed most of the articles were Brazil (27.3%) and USA (18.2%). The studies involved experiments in animals (63.6%) and/or using bacteria or cells (36.4%), no clinical study was found. Ten studies investigated CBD and one study investigated its derivative HU-308. CBD dosages used across the studies varied widely, with 3–10 mg/kg being the most frequent range of dosage in animal studies, with intraperitoneal administration, whereas direct contact of CBD was the mode of administration in cell studies. The presentation of CBD was mainly in the form of a powder dissolved in saline (54.5%). The most frequent manufacturers of CBD were from the USA (36.4%) and Germany (27.3%). Three different applications of CBD in dentistry were observed, namely periodontal therapy (45.4%), aid for bone regeneration processes in oral surgery (27.3%), and a general use in oral therapies (27.3%). Sponsors of the studies were either federal funding agencies, governmental authorities, or universities. We could not detect any studies that were sponsored by private companies.

### 3.2. Characteristics of Patents

Characteristics of the 13 patents included are shown in Table 3 [27,28,29,30,31,32,33,34,35,36,37,38,39]. The inventors were based in only two countries: China (53.8%) and USA (46.2%). One patent was published in 2014 and the others between 2017 and 2020. The main claims presented in the patents involved compositions for oral care (e.g., toothpaste, mouthwash, dental floss), tooth whitening ability, injury repair potential, antifungal and anti-inflammatory agents, analgesic effects, and even treatment of viral infections. One group of inventors and one company from China were the inventors of four patents included. A total of 76.9% of the patents were filed in association with a company.

### 3.3. Synthesis of Results and Summary of Evidence

The main applications of CBD and its synthetic analogs in dentistry are illustrated in Figure 2. The in vitro and in vivo studies included in this review reported that CBD may have beneficial effects as an adjunct to periodontal therapy. When HU-308, a CBD-derivative drug, was tested, the study demonstrated that the CB2 receptor played a role in the control of periodontal damage and the oral alterations it causes in the alveolar bone, gingival tissue, and salivary function [23]. CBD showed anti-inflammatory and anti-bone resorption properties [23]. Modulation of host responses by CBD was described as an interesting and alternative approach to conventional periodontal therapy [18,24]. Furthermore, CBD could promote fibrotic gingival enlargement, increase in the production of gingival fibroblasts and production of healing growth factors. It also led to a decrease in the production and activity of metalloproteinases [24]. The possibility was also observed that CBD may have antimicrobial properties and be effective in reducing the colony count of bacterial strains of dental plaque, which could control and reduce inflammatory periodontal diseases of bacterial origin [22,25].

The anti-inflammatory properties of CBD also made it a suitable alternative for the prevention and treatment of oral mucositis, by reducing the inflammatory process and the severity of lesions [9,19]. CBD was shown to enhance the healing process of common ulcers by reducing the number of colony counts of bacterial strains in dental plaque when compared with well-established synthetic oral care products [25]. Furthermore, CBD alone was shown to improve fracture healing by promoting the stimulation of mesenchymal cells via activation of p42/44 receptors at the lesion site and stimulating their differentiation into osteoblasts, which indicated biocompatibility and osteoinductivity [6,7]. Furthermore, mesenchymal stem cells isolated from dental tissues were capable of differentiating into osteoblasts when exposed to a low CBD dose, and a better expression of bone proteins was observed [26].

## 4. Discussion

This scoping review mapped the available literature and patents on the use of CBD for dentistry-related applications. Most of the peer-reviewed literature suggested that the positive effects of CBD could be attributed to its anti-inflammatory, analgesic, antimicrobial, biological, and osteoinductive properties. The patents also seemed to rely on these properties to offer technological applications of CBD in dentistry. The dental subjects mainly involved in the literature and patents reviewed were periodontology, oral medicine, and oral surgery. These subjects, the potential technological applications of CBD, and future research challenges on the topic are discussed as follows.

### 4.1. Periodontal Therapy

CBD has anti-inflammatory properties and was shown to be capable of reducing alveolar bone loss in induced periodontitis [18,25]. Pharmacologically, the selective activation of CBD at CB2 receptors allows therapeutic, analgesic, or anti-inflammatory effects, while they prevent the secondary effects resulting from the activation of CB1 receptors [5]. These anti-inflammatory properties manifested themselves in a similar way to that of other endogenous cannabinoids, including anandamide. Specifically, the studies demonstrated that anandamide is present and regulated during the progression of periodontal disease and is involved in the suppression of pro-inflammatory mediators. Moreover, it is possibly physiologically involved in the protection of periodontal tissues against excess inflammation [18]. Furthermore, it can block NT-kB, a regulator of the immune and inflammatory response, normally activated by endotoxins from bacterial lipopolysaccharides, and is also capable of reducing the production of inflammatory mediators such as interleukins [5]. Effects also have been shown to involve the inhibition or modulation of the production of cytokines, chemokines, and pro-inflammatory growth factors, along with interference in macrophage and neutrophil migration and reduction of oxidative and nitrosative stress [2]. The anti-inflammatory potential of CBD, similar to that of endocannabinoids, has directly or indirectly demonstrated similar behavior at cannabinoid receptors, which could have interesting clinical implications. However, the results of another study suggested that CBD may promote increased gingival fibrosis, thereby increasing the production of gingival fibroblasts, transformative growth factor b and fibronectin with decrease in matrix metalloproteinase production and activity [24]. Low CBD concentrations have increased transforming growth factor beta levels by as much as 40% at 24 h, suggesting that CBD may promote fibrosis in this way [24]. Moreover, CBD may increase the levels of anandamide, which may promote fibrosis via CB1 or other receptors [24].

### 4.2. Oral Medicine

These anti-inflammatory and analgesic properties of CBD have been reported to be dose-responsive, without ideal doses having been established for a possible antioxidant and anti-inflammatory action [9,21,26]. This combination of anti-inflammatory, antioxidant, and analgesic properties of CBD may render a more potent action than classical antioxidants, which could generate a change in the production of pro-inflammatory mediators responsible for inflammatory pathologies such as oral mucositis [9]. This could make it possible to use of CBD as a potential therapy for the treatment of the symptoms of mucositis as CBD resulted in favorable epithelial changes in ulcer lesions in vivo. An important point capable of influencing this tissue response is the possible ability of CBD to induce effects on keratinocytes without showing side effects. Despite its anti-inflammatory effect, CBD did not accelerate wound healing [19]. Therefore, the influence of CBD on keratinocytes continues to be controversial and further studies are needed to investigate its effects and identify the mechanisms of action.

In an in vivo study [23], the synthetic analog HU-308 caused a reduction in alveolar bone loss and inflammatory mediators in gingival tissues, which are increased by lipopolysaccharide-induced periodontitis in the absence of treatment. The main effect is local and produced by activation of the CB2 receptors, stimulating the differentiation of osteoblast cells and mitigating osteoclastogenesis. This suggests that the signaling of the CB2 receptor produces blockade of bone loss through a direct action on bone cells and simultaneously by inhibiting the expression of pro-reabsorption cytokines [23].

CBD showed effective antimicrobial properties in the reduction of colonies of bacterial strains against oral bacteria and two biofilms [22]. High doses of CBD suppressed the growth of *Porphyromonas gingivalis* and *Filifactor alocis*, which are key components of subgingival microbiota [22]. Dental plaque is a balanced, biofilm-organized structure that contains bacteria responsible for various oral diseases such as gingivitis, periodontitis, and dental caries. The majority of the microorganisms which dental biofilms are composed of are Gram-positive bacteria and could be susceptible to the action of CBD, which showed a greater reduction in bacterial colonies when compared with other oral hygiene products [25]. However, the efficacy of CBD could vary from one individual to another due to the microbial diversity of oral biofilms [25].

### 4.3. Adjuncts to Oral Surgery and Traumatology

CBD has shown favorable biological and osteoinductive properties. When used alone or in combination with other substances, CBD was sufficiently effective and reliable to produce cell migration and bone differentiation [6,7,26] and exerted an impact on the pro-migratory activity of microglial cells through the activation of the endocannabinoid system [25]. Bone cells express cannabinoid receptors and endocannabinoid metabolizing enzymes, and cannabinoid receptors are specifically expressed in skeletal sympathetic nerve endings. Moreover, cannabinoids play an important role in regulation and remodeling of bone mass [7]. Furthermore, studies have shown that CBD increased the expression of PLOD1 gene [6,7,26], which indirectly increases the cross-linking ratio of collagen, showing an indicator of collagen maturity and potential for inducing bone protein expression and enhancing mineralization. Both actions may improve neobone formation and the biomechanical properties of bone tissue. However, there are still questions to be answered about the selective mechanisms of the receptors in the healing and regeneration of bone.

### 4.4. The Scientific and Technological Scenario

In line with the recent literature on the use of CBD in dentistry, our findings also revealed an increasing interest in technology and appropriation relative to dental applications of CBD in recent years. Despite the increasing progress in filing these patents, only one patent [31] dating back to 2018 appeared simultaneously with publication of the results in a scientific study [24] in 2020. The patents deposited with the application of CBD in dentistry were mostly related to products for oral care. This area seems to offer space for exploration of CBD due to the multiplicity of potential dentistry-related applications and the increasing interest in natural products [40,41]. The incentive to the industrial exploitation of CBD in dentistry through the consolidation of policies to stimulate scientific and technological projects could result in economic growth based on research [42]. In spite of the challenges, today CBD and its synthetic analogs are seen as promising compounds in multiple areas, especially because of the low cost of these phyto-derivatives [43].

### 4.5. Limitations

A number of limitations relative to research and technological applications of CBD in dentistry can be cited. One limitation is the absence of clear regulations governing the supervision of the quality of CBD [40], which could lead to varied efficiency of CBD for dental applications across origins of its supply and manufacturers. The constituents of plant-derived natural products vary greatly, often due to variation in environmental conditions. Consequently, the quality and concentration of natural products depend on their geographic location, vegetation, and extraction conditions [44]. Therefore, they may contain different chemical constituents that promote their therapeutic activities. These aspects leave room for research concerning the effect of different constituents of CBD-containing solutions for dental applications. Another limitation highlighted in this review was the heterogeneity of analytical techniques used for characterizing the products. Variations in control of the source of these materials, their storage, production process, and contamination could all be relevant factors influencing quality of products and their therapeutic results [45].

### 4.6. Future Challenges

The scientific articles reviewed indicated challenges to the current clinical use of CBD in dentistry and a potentially extensive field for research and products using CBD and its synthetic analogs, including:Studying of the behavior, dosage, and mechanisms of action of CDB used in dentistry-related clinical conditions;Acceptability of CBD-based treatments among dental patients;Standardization of the methods used for extraction of CBD and standardization of in vitro and in vivo tests with this compound [46];Evaluation of the cytocompatibility of CBD in order to develop safe products;More basic studies are needed to increase safety of CBD for use in dental patients;Since CBD has been approved for use in patients in many countries, clinical studies with patients are also needed in dentistry, which means that the available evidence is of low quality, low certainty, and prone to risk of bias.

## 5. Conclusions

Cannabidiol has been gaining increasing attention in contemporary science and society but seems to have been little explored in dentistry-related research and patents to date. Dental research evidence suggests that CBD has anti-inflammatory, analgesic, antimicrobial, biological, and osteoinductive properties for potential periodontal, oral surgery, and oral medicine applications. Patents available indicate that at present, the interest of industry relies on compositions for oral care products including toothpaste, mouthwash, and dental floss. There appears to be extensive room available for research and technological applications of CBD in dentistry, especially as far as the lack of clinical trials is concerned.

## Figures and Tables

**Figure 1 dentistry-10-00193-f001:**
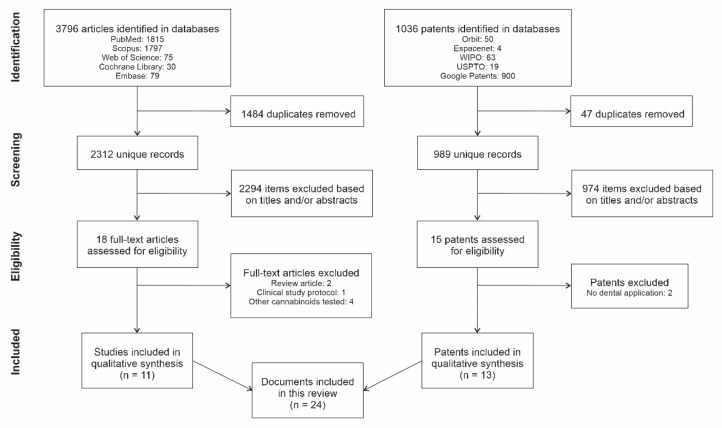
Flowchart of the selection process of articles and patents.

**Figure 2 dentistry-10-00193-f002:**
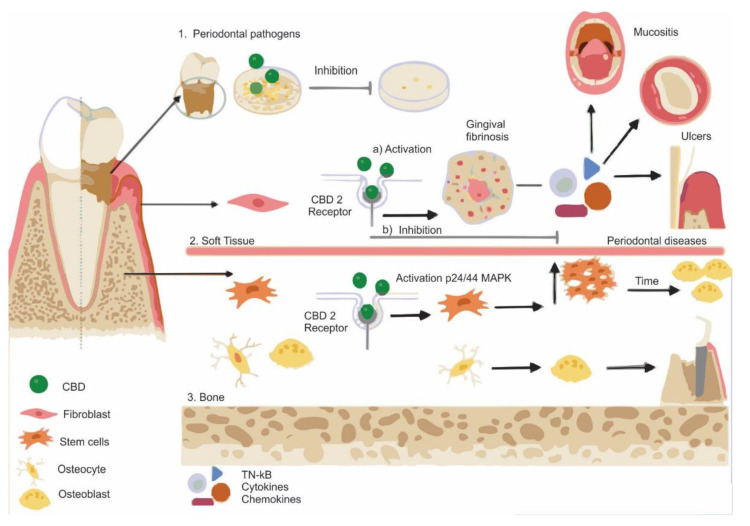
Applications of CBD in dentistry according to the reviewed literature. (1) CBD could reduce colonies of periodontal pathogens, showing antimicrobial properties. (2) In soft tissues, CBD could activate CB2 receptors in fibroblasts and lead to gingival fibrosis by increasing the production of gingival fibroblasts via stimulation of growth factors and decrease in matrix metalloproteinases. Simultaneously, CBD could inhibit or modulate the production of cytokines, chemokines, and pro-inflammatory growth factors, interfering with the migration of macrophages and neutrophils, generating anti-inflammatory, antioxidant, and analgesic potentials. Potential treatments include ulcers, mucositis, and as an adjunct in periodontal diseases. (3) In bone, CBD may promote the stimulation of mesenchymal cells via p42/44 mitogen-activated protein kinases (MAPK) toward the lesion site and its differentiation into osteoblasts.

**Table 1 dentistry-10-00193-t001:** Search strategy used in the different databases.

PubMed/MEDLINE	(THC) OR (Tetrahydrocannab) OR (Dronabin) OR (Cannab) OR (Phytocannab) OR (Marijuana) OR (Synthetic cannab) OR (Cannabidiol) AND (Bone and Bones [Mesh] OR (Bone and Bones) OR (Bones and Bone Tissue) OR (Bones and Bone) OR (Bone Tissue) OR (Bone Tissues) OR (Tissue, Bone) OR (Tissues, Bone) OR (Bony Apophyses) OR (Apophyses, Bony) OR (Bony Apophysis) AND (Dentistry) OR (Implants, Dental) OR (Dental Implant) OR (Implant, Dental) OR (dental Pain) OR (anti-inflammatory) OR (Pulp, Dental) OR (Pulps, Dental) OR (Dental Pulps) OR (Regeneration, Periodontal Guided Tissue) OR (Guided Periodontal Tissue Regeneration) OR (Periodontal Guided Tissue Regeneration) OR (Bone reparation)
Scopus	ALL (“THC” OR “Tetrahydrocannab” OR “Dronabin” OR “Cannab” OR “Phytocannab” OR “Synthetic cannab” OR “Cannabidiol” AND (“Bone and Bones” OR “Bone and Bones” OR “Bones and Bone Tissue” OR “Bones and Bone” OR “Bone Tissue” OR “Bone Tissues” OR “Tissue, Bone” OR “Tissues, Bone” OR “Bony Apophyses” OR “Apophyses, Bony” OR “Bony Apophysis”) AND (“Dentistry” OR “Implants, Dental” OR “Dental Implant” OR “Implant, Dental” OR “dental Pain” OR “anti-inflammatory” OR “Pulp, Dental” OR “Pulps, Dental” OR “Dental Pulps” OR “Regeneration, Periodontal Guided Tissue” OR “Guided Periodontal Tissue Regeneration” OR “Periodontal Guided Tissue Regeneration” OR “Bone reparation”)
Web of Science	TS = (THC) OR (Tetrahydrocannab *) OR (Dronabin *) OR (Cannab *) OR (Phytocannab *) OR (Marijuana *) OR (Synthetic cannab *) AND (Dentistry) OR (Implants, Dental) OR (Dental Implant) OR (Implant, Dental) OR (dental Pain) OR (anti inflamatory) OR (Pulp, Dental) OR (Pulps, Dental) OR (Dental Pulps) OR (Regeneration, Periodontal Guided Tissue) OR (Guided Periodontal Tissue Regeneration) OR (Periodontal Guided Tissue Regeneration) OR (Bone reparation)
Cochrane Library	(THC) OR (Tetrahydrocannab) OR (Dronabin) OR (Cannab) OR (Phytocannab) OR (Marijuana) OR (Synthetic cannab) OR (Cannabidiol) AND (Bone and Bones) [Mesh] OR (Bone and Bones) OR (Bones and Bone Tissue) OR (Bones and Bone) OR (Bone Tissue) OR (Bone Tissues) OR (Tissue, Bone) OR (Tissues, Bone) OR (Bony Apophyses) OR (Apophyses, Bony) OR (Bony Apophysis) AND (Dentistry) OR (Implants, Dental) OR (Dental Implant) OR (Implant, Dental) OR (dental Pain) OR (anti-inflammatory) OR (Pulp, Dental) OR (Pulps, Dental) OR (Dental Pulps) OR (Regeneration, Periodontal Guided Tissue) OR (Guided Periodontal Tissue Regeneration) OR (Periodontal Guided Tissue Regeneration) OR (Bone reparation)
Embase	(THC) OR (Tetrahydrocannab) OR (Dronabin) OR (Cannab) OR (Phytocannab) OR (Marijuana) OR (Synthetic cannab) OR (Cannabidiol) AND (Bone and Bones) [Mesh] OR (Bone and Bones) OR (Bones and Bone Tissue) OR (Bones and Bone) OR (Bone Tissue) OR (Bone Tissues) OR (Tissue, Bone) OR (Tissues, Bone) OR (Bony Apophyses) OR (Apophyses, Bony) OR (Bony Apophysis) AND (Dentistry) OR (Implants, Dental) OR (Dental Implant) OR (Implant, Dental) OR (dental Pain) OR (anti inflamatory) OR (Pulp, Dental) OR (Pulps, Dental) OR (Dental Pulps) OR (Regeneration, Periodontal Guided Tissue) OR (Guided Periodontal Tissue Regeneration) OR (Periodontal Guided Tissue Regeneration) OR (Bone reparation)
Orbit	THC OR Tetrahydrocannab OR Dronabin OR Cannab OR Endocannab OR Phytocannab OR Marijuana OR Synthetic cannab OR Cannabidiol OR Endocannabinoide AND “Bone and Bones OR Bones and Bone Tissue OR Bones and Bone OR Bone Tissue OR Bone Tissues OR Tissue, Bone OR Tissues, Bone OR “Bony Apophyses OR Apophyses, Bony OR Bony Apophysis OR Dentistry OR Implants, Dental OR Dental Implant OR Implant, Dental OR dental Pain OR anti-inflammatory OR Pulp, Dental OR Pulps, Dental OR Dental Pulps OR Regeneration, Periodontal Guided Tissue OR Guided Periodontal Tissue Regeneration OR Periodontal Guided Tissue Regeneration or Bone reparation
Google patents	THC OR Tetrahydrocannab OR Dronabin OR Cannab OR Endocannab OR Phytocannab OR Marijuana OR Synthetic cannab OR Cannabidiol OR Endocannabinoide AND “Bone and Bones OR Bones and Bone Tissue OR Bones and Bone OR Bone Tissue OR Bone Tissues OR Tissue, Bone OR Tissues, Bone OR “Bony Apophyses OR Apophyses, Bony OR Bony Apophysis OR Dentistry OR Implants, Dental OR Dental Implant OR Implant, Dental OR dental Pain OR anti-inflammatory OR Pulp, Dental OR Pulps, Dental OR Dental Pulps OR Regeneration, Periodontal Guided Tissue OR Guided Periodontal Tissue Regeneration OR Periodontal Guided Tissue Regeneration or Bone reparation
USPTO	THC OR Tetrahydrocannab OR Dronabin OR Cannab OR Endocannab OR Phytocannab OR Marijuana OR Synthetic cannab OR Cannabidiol OR Endocannabinoide AND “Bone and Bones OR Bones and Bone Tissue OR Bones and Bone OR Bone Tissue OR Bone Tissues OR Tissue, Bone OR Tissues, Bone OR “Bony Apophyses OR Apophyses, Bony OR Bony Apophysis OR Dentistry OR Implants, Dental OR Dental Implant OR Implant, Dental OR dental Pain OR anti-inflammatory OR Pulp, Dental OR Pulps, Dental OR Dental Pulps OR Regeneration, Periodontal Guided Tissue OR Guided Periodontal Tissue Regeneration OR Periodontal Guided Tissue Regeneration or Bone reparation
WIPO	THC OR Tetrahydrocannab OR Dronabin OR Cannab OR Endocannab OR Phytocannab OR Marijuana OR Synthetic cannab OR Cannabidiol OR Endocannabinoide AND “Bone and Bones OR Bones and Bone Tissue OR Bones and Bone OR Bone Tissue OR Bone Tissues OR Tissue, Bone OR Tissues, Bone OR “Bony Apophyses OR Apophyses, Bony OR Bony Apophysis OR Dentistry OR Implants, Dental OR Dental Implant OR Implant, Dental OR dental Pain OR anti-inflammatory OR Pulp, Dental OR Pulps, Dental OR Dental Pulps OR Regeneration, Periodontal Guided Tissue OR Guided Periodontal Tissue Regeneration OR Periodontal Guided Tissue Regeneration or Bone reparation
Espacenet	THC OR Tetrahydrocannab OR Dronabin OR Cannab OR Endocannab OR Phytocannab OR Marijuana OR Synthetic cannab OR Cannabidiol OR Endocannabinoide AND “Bone and Bones OR Bones and Bone Tissue OR Bones and Bone OR Bone Tissue OR Bone Tissues OR Tissue, Bone OR Tissues, Bone OR “Bony Apophyses OR Apophyses, Bony OR Bony Apophysis OR Dentistry OR Implants, Dental OR Dental Implant OR Implant, Dental OR dental Pain OR anti-inflammatory OR Pulp, Dental OR Pulps, Dental OR Dental Pulps OR Regeneration, Periodontal Guided Tissue OR Guided Periodontal Tissue Regeneration OR Periodontal Guided Tissue Regeneration or Bone reparation

*: The asterisk is used for searching terms with other variations at the end of the word.

**Table 2 dentistry-10-00193-t002:** Characteristics of the original articles included, n = 11.

First Author, Year	Journal	Country	Study Type	CBD or Analog	Dosage	Administration	Presentation	Manufacturer	Main Application	Sponsor
Napimoga, 2009 [18]	Int. Immunopharmacol.	Brazil	Animal	CBD	1 mg/kg	Injected i.p	CBD powder dissolved in 2% polysorbate 20/Tween 80- in saline	THC Pharm, Germany	Periodontal therapy	CAPES and UNIUBE, Brazil
Klein, 2018 [19]	Phytother. Res.	Brazil	Animal	CBD	5–10 mg/kg	Injected i.p	CBD powder dissolved in 2% polysorbate 20/Tween 80- in saline	THC Pharm, Germany	Oral therapy	CAPES and PUCRS, Brazil
Kogan, 2015 [8]	J. Bone Miner. Res.	Israel	Animal	CBD	5 mg/kg	Injected i.p	CBD powder dissolved in ethanol/emulphor/saline	MAX-IV laboratory, Sweden	Aid for bone regeneration	NIH, USA and Israel Anti-Drug Authority
Kamali, 2019 [7]	Mater. Sci. Eng. C Mater. Biol. Appl.	Iran	Animal	CBD	30 mg/kg	Direct contact	CBD powder dissolved in 2% polysorbate 20/Tween 80- in saline	Tocris company, USA	Aid for bone regeneration	Shiraz University, Royan Institute, and the Iran National Science Foundation, Iran
Cuba, 2017 [20]	J. Clin. Pharm. Ther.	Brazil	Animal	CBD	3, 10, 30 mg/kg	Injected i.p	CBD powder dissolved in Tween 80- in saline	THC Pharm, Germany	Oral therapy	CAPES and PUCRS, Brazil
Scionti, 2016 [21]	Front. Physiol.	Italy	Animal	CBD	10 mg/kg	NS	Pure CBD (>99%)	Greenhouse cultivation at CREA-CIN, Italy	Periodontal therapy	Health Ministry, Italy
Gu, 2019 [22]	Front. Immunol.	USA	Animal	CBD	10 mg/kg	NS	NS	Cayman Chemical Co., USA	Periodontal therapy	NIDCR, USA
			Cells		0.1–1.0 µg/mL	Direct contact	NS			
Ossola, 2016 [23]	J. Periodontol.	Argentina	Animal	HU-308	200 µL per tooth	Topical	HU-308 powder dissolved 100% ethanol/saline	Tocris, USA	Periodontal therapy	University of Buenos Aires and CONICET, Argentina
Rawal, 2012 [24]	J. Periodontal. Res.	USA	Cells	CBD	0.01–30 µM	Direct contact	CBD powder dissolved in methanol	Sigma-Aldrich, USA	Periodontal therapy	University of Tennesse, USA
Stahl, 2020 [25]	Cureus	Belgium	Cells/bacteria	CBD	NS	Direct contact	NS	NS	Oral therapy	NS
Petrescu, 2020 [26]	Medicina	Romania	Cells	CBD	0.75 µM	Direct contact	NS	NS	Aid for bone regeneration	CNCS, Romania

CBD: cannabidiol; i.p: intraperitoneal; NS: not specified; CAPES: Coordination for the Improvement of Higher Education Personnel; UNIUBE: University of Uberaba; PUCRS: Pontifical Catholic University of Rio Grande do Sul; NIH: National Institutes of Health; NIDCR: National Institute of Dental and Craniofacial Research, USA; CONICET: National Scientific and Technical Research Council; CNCS: National Council of Scientific Research.

**Table 3 dentistry-10-00193-t003:** Characteristics of the patents included, n = 13.

Patent Number(s)	Country	Year	Title	Main Claims	Inventors	Company
US20190076349 US10172786 EP3233021 EP3233021 WO2016/100516 US20160166498	USA	2014	Oral care composition comprising cannabinoids	Toothpaste, tooth powder, or mouthwash solution	George Anastassov, Lekhram Changoer	Axim Biotechnologies
CN109939012	China	2017	Use of the tooth whitening CBD	Toothpaste, dentifrice, mouthwash, gel, gum or film-forming agent for tooth whitening	Zhang Ke, Tan Xin, Yu Zhaohui, Lian Meng, Chang Tanran, Jin Qian	Han Yi Biotechnology
CN109939011	China	2017	Composition comprising a cannabinoid toothpaste and its preparation method	Toothpaste with oral injury repair potential	Zhang Ke, Tan Xin, Yu Zhaohui, Lian Meng, Chang Tanran, Jin Qian	Han Yi Biotechnology
WO2019/055420 US20190076343	USA	2017	Oral care formulations and methods for use	Toothpaste, mouthwash, chewing gum, lozenge, coated interdental device, and coated dental floss as oral care products	Gerald Curatola	NS
WO2019030762A2 EP3664795A2 US20200222361A1	USA	2018	Cannabis and derivatives thereof for the treatment of pain and inflammation related with dental pulp and bone regeneration related to dental jaw bone defects	Composition for treating dental pulp inflammation/infection and bone defects	Veronica Stahl	CannIBite
CN109939011A	China	2019	A kind of toothpaste and preparation method thereof containing cannabinoid	Toothpaste with oral injury repair potential	Zhang Ke, Tan Xin, Yu Zhaohui, Lian Meng, Chang Tanran, Jin Qian	Han Yi Biotechnology
CN109939012A	China	2019	Application of the cannabidiol in tooth whitening	Tooth whitening product	Zhang Ke, Tan Xin, Yu Zhaohui, Lian Meng, Chang Tanran, Jin Qian	Han Yi Biotechnology
CN109925246A	China	2019	A kind of toothpaste and preparation method thereof with sterilization, anti-inflammatory, and analgesic efficacy	Toothpaste	Luan Yunpeng, Zheng Shuangqing, Li Zhipeng	NS
US20200163746A1	USA	2019	Cannabinoid-infused post-operative dental dressing	Dental dressing pouch for an oral surgical site	David Tumey, Harry Panahi	NS
WO2019239357A1	USA	2019	Dental care cannabis device and use thereof	Device for topical administration	Dana Zeitouni, Aharon M. Eyal	Buzzelet Development and Technologies
CN111671698A	China	2019	Cannabinoid-containing functional toothpaste	Toothpaste with antifungal, anti-inflammatory, analgesic, and neuroprotector effects	Xu Haiyan, He Wei, Xia Meilian, He Zhijiang, Kite Xiao, Xiao Hong, Xie Lingyu	Jiangxi Caoshanhu Oral Care Products
CN111973480A	China	2019	Oral care composition and application thereof	Toothpaste, dentifrice, mouthwash, oral spray, oral patch, chewing gum or buccal tablet for oral hygiene	Chen Qi, Xiao Fengsha, Gull Lianmeng, Chang Tanran, Li Ruyan, Li Qingzhong	Yunnan Hanmeng Pharmaceutical
WO2020257588A1	USA	2020	Compositions for preventing and treating viral infections	Composition for treating viral infections	Maria Crisler, Emma Diponio, Philip Kennedy	Shaman Naturals

Abbreviations: i.p: intraperitoneal, NS: not specified.

## Data Availability

The data presented in this study are openly available in Open Science Framework at https://doi.org/10.17605/OSF.IO/YK65C (accessed on 10 October 2022).

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
