# Peer review of "Cannabidiol in Dentistry: A Scoping Review"

_dentistry, 2022, doi:10.3390/dj10100193_

Round 1

Reviewer 1 Report

Thank you for submitting: Cannabidiol in dentistry: A scoping review.

Abstract: provides enough information about the paper

Keywords: ok

Introduction: broad enough to talk about the current use of cannabidiol.

Materials and methods: Why did you chose OSF registries instead other like PROSPERO? It is ok to put it on OSF. Ok to the question. Why did you excluded case reports, case series or review?

It is perfect you explain who did the research. A lot of databases were consulted.

Characteristics of studies: Maybe you could graph the results of countries, the studies involved experiments… to make it more visual.

Figure 2 is own elaboration?

Discussion is ok, limitations and future challenges are specified.

Conclusions are ok.

Author Response

Thank you for your review and comments. A few notes:

  • OSF was used because PROSPERO does not accept protocols for Scoping Reviews (https://www.crd.york.ac.uk/prospero/)
  • Explanation for the exclusions were added to the revised manuscript
  • Figure 2 is own elaboration
  • We tried graphing the results by country but the image did not offer much information because the papers were scattered along the globe and the patents concentrated in two countries

Reviewer 2 Report

manuscript is written in simple and appropriate way.

Author Response

Thank you for your review.

Reviewer 3 Report

The manuscript aims to summarize the state-of-art of cannabidiol in the dental field. The topic is very interesting within the scope of the journal. The search strategy in different databases makes the results reasonable and scientific. The figures are visible to the readers, The manuscript can be considered for publication. There are only a few comments:

1. the state-of-art of cannabidiol should be further explained in the introduction.

2. Limitations and future challenges should be separated. Also, the outlook and future challenges should be written point-to-point, which is better for the readers.

2. English language and style could be a minor spell check required.

Author Response

Thank you for your review and comments. A few notes:

  • The state-of-the art of CBD was further explored in the Intro chapter
  • Limitations and future challenges were separated, as suggested. In addition, the future challenges was written point-to-point (we agree it is more reader friendly now)
  • The text was checked for spelling errors

Reviewer 4 Report

Dear authors,

Congratulations on your work!

Cannabidiol in dentistry: A scoping review, is a well-documented paper. 

The aim of this scoping review was to map the contemporary scientific and technological scenarios related to the use and applications of CBD in dentistry at present.

The research for articles was carried out in five international databases.

All records identified were imported into the EndNote program

Characteristics of the 11 original studies included are presented

Characteristics of the 13 patents included are shown 

Figure 2. Applications of CBD in dentistry is of good quality

Discussion is comprehensive

Conclusion is fair

References are adequate

Author Response

Thank you for your review and comments.